# Research Trends in Crop–Livestock Systems: A Bibliometric Review

**DOI:** 10.3390/ijerph19148563

**Published:** 2022-07-13

**Authors:** Guoting Yang, Jing Li, Zhen Liu, Yitao Zhang, Xiangbo Xu, Hong Zhang, Yan Xu

**Affiliations:** 1Key Laboratory of Ecosystem Network Observation and Modeling, Institute of Geographical Science and Natural Resources Research, Chinese Academy of Sciences, Beijing 100101, China; ygt19834556613@163.com (G.Y.); ytzhang1986@163.com (Y.Z.); ydxu.ccap@igsnrr.ac.cn (X.X.); 2Rural Energy and Environment Agency, Ministry of Agriculture and Rural Affairs, Beijing 100125, China; zhanghong@sxu.edu.cn; 3Yellow River Delta Modern Agricultural Engineering Laboratory, Chinese Academy of Sciences, Beijing 100101, China; liuzhen@igsnrr.ac.cn; 4International Ecosystem Management Partnership, United Nations Environment Programme, Beijing 100101, China; 5University of Chinese Academy of Sciences, Beijing 100049, China

**Keywords:** agro-pastoral systems, CiteSpace, ecological benefits, economic benefits

## Abstract

Agricultural straw waste and livestock manure are often misplaced resources. The separation of planting and breeding has induced significant pressure on the environment. Thus, there is a growing need for a system that can integrate crop and livestock farming to improve resource efficiency. To clarify the current state of crop–livestock systems in China and elsewhere, a bibliometric analysis was conducted for a total of 18,628 published English and 3460 published Chinese research articles and dissertations on circular agriculture. The published research papers were taken from the ISI Web of Science and CNKI database to explore research hotpots, research methods, theme trends, and mainstream technical models of crop–livestock systems from 1981 to 2021. Recent progress in crop–livestock systems was analyzed from ecological, economic, social benefits, and stakeholder perspectives. The research results showed that compared with traditional agricultural models, crop–livestock systems had far more ecologic and social benefits, including gaining higher net income and input-output ratio, improving soil quality, and mitigating global warming. The drivers of crop–livestock systems’ development were also analyzed from stakeholders’ perspectives. The study provides insights into the development of circular agriculture by reducing the pollution risks of agricultural waste and improving both ecological and economic benefits of the system.

## 1. Introduction

Developing crop–livestock systems is an effective way to accelerate the pace of green agriculture reform and improve the quality of farm products which is the basis of food security. In crop–livestock systems, the crop systems provide the feed for the livestock farms, which in turn produce manure fertilization for plants. This is a “conversion cycle” of materials and energy between the crop industry and the livestock industry, which promotes internal resource recycling of the system [1]. With the development of Agricultural mechanization and specialization, livestock farms tend to industrialize [2]. However, agricultural specialization often leads to the separation of the crop from livestock [3], significantly reducing the utilization rate of natural resources and causing environmental problems such as greenhouse gas emissions and nitrate pollution [4,5]. At the same time, in the face of the challenges of climate change, farmers are well aware of the value of wastewater as a kind of fertilizer instead of the impact of contaminated water on crops. Furthermore, smallholders face more threats than well-settled farmers, which means a new type of ecological agriculture system is now urgently needed [6,7,8]. By combining crop farming and animal husbandry, crop–livestock systems improve nutrient cycling and energy efficiency, thus gaining benefits economically and ecologically [9]. Crop–livestock systems can also contribute to the achievement of carbon emission reduction targets and help cope with the negative impact of monetary policy uncertainty by increasing the amount of renewable energy [10,11].

Research on crop–livestock systems ranges from qualitative to quantitative studies and from processes to mechanisms. However, there is no available systematic study to quantitatively analyze the performances of crop–livestock systems and traditional agricultural models in terms of net income, input-output ratio, soil quality, and their effects on mitigating global warming. Moreover, there is little systematic analysis concerning the mechanisms influencing crop–livestock systems. Bibliometrics is the quantitative analysis that shows the internal structures and development trends of certain research fields [12]. Currently, it has been widely used in ecology [13], soil science [14], environmental science [15], geography [16], and other research fields and provided meaningful findings. However, no bibliometric research has been conducted on crop–livestock systems in China or elsewhere in the world.

To provide an in-depth overview of crop–livestock systems, the CiteSpace and Vos Viewer were used to analyze and visualize the current situation and development trend of crop–livestock systems. The objectives of the study were to: (i) summarize the popularity and methods of research and the models of thematic trends and mainstream techniques of crop–livestock systems in the past 40 years; (ii) quantitatively analyze and compare economic and ecological benefits of crop–livestock systems and traditional agriculture; and (iii) systematically analyze the social benefits of crop–livestock systems from stakeholders’ perspective. The results of the study may provide insights for lowering the pollution risk of agricultural waste, improving ecological and economic benefits of crop–livestock systems, and further developing circular agriculture.

## 2. Data Source

Based on the ISI Web of Science (WOS) and CNKI database, Chinese and English literature were retrieved by the keywords as TS = “circular farming or circular farm or mixed farming or mixed farm or crop–livestock system or circular agriculture or mixed agriculture or planting and breeding”. Data concerning agricultural and animal husbandry systems during the period of 1981–2021 were collected on 13 November 2021. Irrelevant records in the field of artificial farmers, mixed farming, farming systems, agricultural systems, animal husbandry, and circular economy were filtered out. After filtering for deduplication by CiteSpace, 16,631 and 5379 valid documents were screened in the WOS and CNKI databases, respectively.

## 3. Crop–Livestock Systems Research

### 3.1. Overall Literature Analysis

The international crop–livestock systems research from 1981 to 2021 (Figure 1) could be divided into three stages. The first stage was the initial development stage (1981–2001), with a total of 2893 papers published (144 papers per year on average), which had a low level of development. The development of international agro-pastoral circular agriculture can be traced back to the dawn of organic agriculture [17]. It was the period when the crop–livestock system began to take shape, since the concept, thought, and practice of organic agriculture emerged, and the impacts of combining agriculture and animal husbandry on soil fertility and the environment attracted great attention. In 1981, the American scholar, William Lockeretz, conducted research in the Midwest of the United States on crop production without the use of modern fertilizers and pesticides, finding that organic methods consumed less fossil energy and reduced soil erosion. This laid the basis for modern crop–livestock systems [18].

The second stage was the stable growth stage (2001–2010), with a total of 3896 publications. The rate of publication during this stage was 354 papers per year (some 2.46 times higher than the first stage), and the development rate rose steadily. Crop–livestock system research was more detailed and comprehensive in terms of the focus on soil health. It ensured environmental sustainability by monitoring the impact of crop–livestock systems on soil carbon and nitrogen storage and optimizing cropping systems. It also explored the sustainable development of crop–livestock systems under no-tillage cultivation, conservation agriculture, and food production.

The final stage was the rapid growth stage (2010–2021), with a total of 12,653 papers published (an average of 1054 per year) and an overall rapid upward trend. Research on crop–livestock systems focused on sustainable grain yield and profitability and venturing into institutional reforms and breakthroughs.

The CNKI literature database could be divided into three stages as well, based on the average annual publication volume. The initial stage (1981–2001) had a total of 255 papers (an average of 12 papers per year) and a slow growth rate. In 1985, Zhang Yuanhao proposed the definition of “circular agriculture” for the first time, arguing that agricultural production required a combination of material circulation and energy conversion. During this period, scholars began to explore the practices and models of crop–livestock systems [19], making preliminary progress in domestic crop–livestock system research.

There was a rapid development stage during 2001–2010, when a total number of 1170 papers were published (an average of 106 papers per year), characterized by a rapid development rate. Research in this period focused on the cost and policies regarding crop–livestock systems, with a significant increase in studies on circular economy and sustainable development.

The third stage was the fluctuating stage (2010–2021), with a total of 4332 publications (an average of 396 papers per year) and fluctuating development. Research in this period focused on agricultural non-point source pollution, manure management, and a low-carbon economy. Sustainable development policies were further strengthened, and circular farms were heading to green production and ecological development.Figure 1Annual distribution of crop–livestock system research for the period 1981–2021.
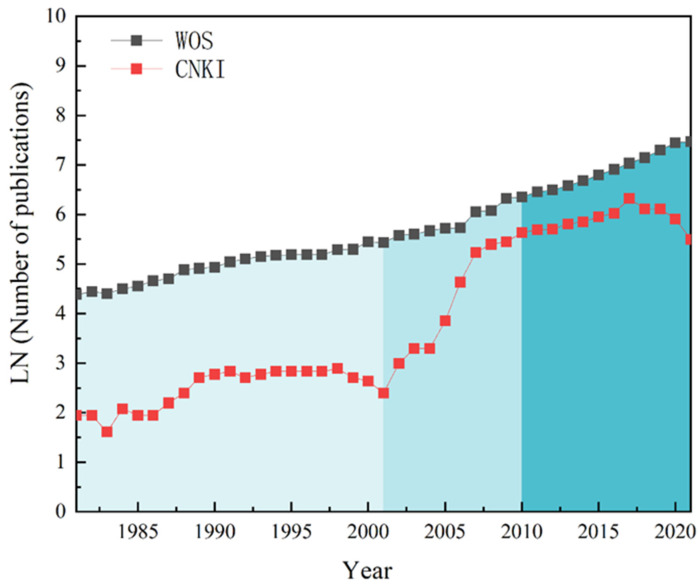


### 3.2. Annual Distribution of Crop–Livestock System Research for the Period 1981–2021

The top ten countries in the WOS document in the field of crop–livestock systems were the United States (4041 papers), China (2201 papers), Germany (849 papers), Australia (848 papers), Brazil (834 papers), the United Kingdom (627 papers), Spain (619 papers), Italy (618 papers), Canada (596 papers), and Japan (559 papers) (Figure 2). The number of published articles is 559–4041, with an in-between centrality of 0.01–1.05. The United States has the most publications among all countries, accounting for 24.30% of the total publications. The core authors of the top five publications were mostly Brazilian scholars. Brazil had formed an academic network centered on Paulo Césarde Faccio Carvalho. The authors were mainly from the Federal University of Rio Grande do Sul, the Federal University of Pará, and the University of Sao Paulo in Brazil. The core American authors, Jason A and Hubbart from West Virginia University had the fourth-largest number of papers among core scholars. Domestic scholars had formed an academic circle led by Weng Boqi, Xi Bin, and Zhou Feng. They belonged to the Key Laboratory of Agro-ecological Processes in the Red Soil Mountains of Fujian Province, the Institute of Agricultural Resources and Agricultural Zoning of the Chinese Academy of Agricultural Sciences, and the School of Urban and Environment of Peking University, respectively.

## 4. Crop–Livestock Systems Research Hotspot

### 4.1. Keyword Analysis

To explore research hotspots in crop–livestock system, a keyword cluster map was drawn, summarizing important research directions in crop–livestock systems (Figure 3). Keywords in the field of agronomy focusing on crop–livestock systems included crop rotation, cover crop, no-tillage, etc., in the WOS database during 1981–2021. Keywords concerning the ecological effects in crop–livestock systems included soil fertility, soil quality, nitrogen, greenhouse gas emission, biodiversity, etc. Keywords concerning economic effects of crop–livestock systems included productivity, yield, sustainability, management, etc. Keywords concerning technologies in circular agriculture in the CNKI database during 1981–2021 included organic fertilizer, biogas engineering, recycling, waste recycling, comprehensive straw utilization, etc. Keywords concerning research methods were life cycle assessment, emergy analysis, structural equation model, analytic hierarchy, etc. Keywords in ecological effects were soil fertility, soil quality, ecological cycle, etc. Then keywords in economic effects were agriculture structure adjustment, input, output, etc.

### 4.2. Crop–Livestock System Models

A crop–livestock system is built on the principle of “reduction, reuse and recycling” [19]. Models in other countries mainly included low-input sustainable models and resource-recycling models [20]. Low-input sustainable models used the no-tillage methods or new biological technology for various soils and crops based on specific conditions. By optimizing the input of agricultural resources and lowering doses of chemical fertilizers and pesticides, the highest possible yields could be acquired with maximized economic benefits [1]. For example, under no-tillage management, corn kernels and straws increased by 157 g.m^−2^ and 80 g.m^−2^, respectively. Soybean grains and straws increased by 72 g.m^−2^ and 122 g.m^−2^, respectively [21]. Based on the transformation between materials and energy in an ecological chain, the resource-recycling models could turn agricultural wastes into new resources [22]. Specifically, “cattle–corn/soybean”, “rice–duck”, or “pig–grass” systems were good models that had been fully verified.

For livestock and poultry manure treatment technologies, China’s Ministry of Agriculture divided the modes of crop–livestock systems into “sewage and fertilizer utilization”, “full manure collection and return to field”, and “specialized energy manure use”. The “sewage fertilizer use” refers to the harmless disposal of aquaculture sewage through multi-stages of sedimentation tanks, oxidation ponds, or biogas projects, and then leftovers are used as base fertilizer after crop harvest or before sowing. This model is the most common and widely used in all regions of China. On this basis, four mainstream models in China’s crop–livestock systems were put forward [23].

(i)The ecological model of “pig, biogas and fruit” in South China uses biogas as the connecting link between livestock/poultry farming and fruit, grain, and vegetable cropping. Biogas digesters, pig houses, and orchards are organically integrated through bioenergy conversion technology [24].(ii)The northern “four-in-one” ecological model integrates a solar greenhouse, livestock/poultry house, and a toilet/biogas digester into a single facility for the purpose of combining organic farming and breeding [25].(iii)The “Five-support” ecological agriculture model in Northwest China combines orchards, rainwater collection facilities, biogas systems, solar pig pens, and toilets into a farming and grazing unit to link swamp, fruit, and grazing into a mutually-reinforcing unit or virtuous cycle of development [25].(iv)The production and life cycle chain mode in the plain area [23], such as the grain-fruit, grain-vegetable, grain-tea, and other patterns. Livestock/poultry manure is treated by bio-gasification to generate organic fertilizers, which are used for planting flowers, grains, trees, and feeding fish and pigs.

The “full manure collection” model for field utilization mainly relies on specialized manure collection and fertilization enterprises to collect manure, store it in oxidation ponds, and treat it into harmless products. Professional fertilization machinery is used after crop harvest and before sowing. The application is concentrated, and the amount of chemical fertilizer used is reduced. This mode is mainly operated in the North China Plain and Northeast China. The “specialized energy manure utilization” model mainly relies on large-scale farms or third-party manure treatment companies to collect manure, process it through large-scale biogas projects or biological natural gas projects, and produce organic fertilizers from biogas residues. Biogas slurry is used in farmland or condensed to enhance its value, and this mode is mainly distributed in the northeast regions, eastern coastal regions, central and eastern regions, North China Plain, and the northwest regions.

### 4.3. Method and Practice of Crop–Livestock System

The data sources for economic, ecological, and social benefits of crop–livestock systems can be roughly divided into statistical and continuous data, survey, and case studies. Representative databases include the China Family Survey (Rural Development) database [26], China Family Panel Studies, CFPS [27,28], and the Agricultural and Rural Survey Database of the Institute of Geographic Sciences and Natural Resources Research [29]. In terms of ecological benefits, emergy analysis, and life cycle assessment are the most frequently used methods to quantitatively evaluate the ecological benefits of crop–livestock systems.

The emergy analysis method uses solar energy value as a unified unit. It has the advantage of facilitating the transformation of economic and ecological flow in the system. It can also be used to quantitatively evaluate the structural and functional characteristics and sustainable development of a crop–livestock system. The life cycle can quantitatively analyze and evaluate the impact of crop–livestock system processes on the environment. In terms of economic benefits, non-parametric data envelopment analysis and parametric stochastic frontier analysis are two mainstream methods to evaluate the economic efficiency of agricultural and animal husbandry circular agriculture. Data envelopment analysis can be used to evaluate the relative efficiency of multiple decision-making processes on the output and input units of crop–livestock systems. Stochastic frontier analysis is used to evaluate the economic efficiency of crop–livestock systems and their influencing factors. In terms of social benefit, the linear regression method can be used to quantitatively analyze the factors driving farmers to apply in crop–livestock systems (Table 1).

#### 4.3.1. Economic Benefits

The economic benefits of crop–livestock systems are mainly reflected by the increase in net income and production ratio. Crop–livestock systems mainly reduce cost input by replacing chemical fertilizers with organic fertilizers, biogas slurry with fuel, and using straw as feed, thus improving economic benefits. In the “rice–duck” circular farming model, duck manure was used as a fertilizer, saving an average of 92.58–161.72 USD/ha in fertilizer costs [36,37], and the cost of pesticides reduces by 58.82% [36,37]. The net income and input–output ratio were 1.91–2.09 times and 1.14–2.12 times that of traditional independent planting models, respectively [36,37].

The three-dimensional farming and animal husbandry cycle modes such as “mountain three-dimensional cropping and breeding” and “litchi-chicken” made full use of space to improve land utilization. Part of the livestock and poultry feed came from crop straw, weeds, and insects within the system, reducing feed input by 29.26% [19,25,38]. Furthermore, the organic fertilizer input into the system was 1133. 59 USD/ha lower than the traditional model [25,38], and the input–output ratio was 1.33–3.35 times that of the traditional model [25,38].

In the “cow–biogas–vegetable” cycle mode, “pig–biogas–maize” cycle mode, “fruit (grass)–pig–biogas–cellar” five-support cycle modes, and similar others, biogas replaced coal for power generation, manure and biogas slurry replaced chemical fertilizers and by doing so, reducing dependency on external resources. In this mode, the proportion of feed cost for cattle and pigs was reduced respectively by 6.79% [25,39,40] and 22.47% [25,39,40], compared with the traditional farming mode. Each m^3^ biogas digester could generate 477.82 kW·h, for an electricity cost of 0.52 RMB/kW·h; saving an electricity cost of 918.88 RMB [19]. Biogas slurry and biogas residue were used in orchards or food crops as organic fertilizers, reducing the amount of chemical fertilizers by 14.61–30% [41,42], with an average annual saving on fertilizer of 1619.24 USD/ha [25,43]. In addition, the agricultural and animal husbandry cycle mode also promotes crop yield and shortens the livestock release time. Swine manure, instead of chemical fertilizers, increased rice production by 15.87%, and the daily gain of pigs fed with biogas slurry was 0.6–0.7 kg. This made it possible to release pigs in advance by about 20 days [24]. The net income of crop–livestock systems with biogas as the link were, respectively, 1.01–4.12 times [25,43,44,45] and 1.24–2.31 [25,43,44,45] times that of the traditional farming model and traditional planting model. The ratio of production to investment are, respectively, 1.03–3.35 times [25,43,44,45] and 1.36–2.05 times of the traditional farming model and the traditional planting model [25,43,45].

In most cases, the net income and production-to-investment ratio of crop–livestock systems were higher than those of the traditional breeding and planting model. However, due to the impact of market price fluctuation on livestock products, the net income and input-output ratio of crop–livestock systems were also lower than those of traditional farming and animal husbandry cycle [40].

#### 4.3.2. Ecological Benefits

The ecological and environmental benefits are mainly reflected by the improvement in soil quality, reduction in environmental pollution, and mitigation of global warming. The process of returning crop–livestock manure to the field changes the physical properties and nutrients of the soil, such as decreasing soil bulk density and increasing soil porosity. In addition, soil organic matter, soil-available phosphorus, and soil-available potassium will be increased; thus, soil quality will be improved. Furthermore, replacing chemical fertilizers with an organic fertilizer in a crop–livestock system reduces greenhouse gas (N_2_O and CO_2_) emissions, slowing down the potential global warming.

With the amendment by applying biogas slurry or manure, soil bulk density decreased by 0.11–0.16 g.cm^−3^ [24,46], and soil porosity increased by 5–20% [24,46] in crop–livestock systems. Soil fertility was also improved by returning manure to the field. Compared with the conventional agricultural model, soil organic matter increased by 0.73–6.05 g.kg^−1^ [24,46]. Soil organic matter in “rice–duck” systems had the greatest increase proportion at 24–65%, while in other crop–livestock systems, SOM increased by 3–31%. Soil-available phosphorus increased by 1.34–18.35 mg.kg^−1^ and available potassium by 2–89% [24,46]. Comparing to other farming modes, soil nutrients increased more significantly under the “rice–duck” model. This is mainly because duck droppings served as fertilizers in paddy fields, among which duck manure was a high-quality organic fertilizer [47]. The roaming and feeding of ducks in rice fields accelerated the return of old and weak rice leaves to the soil, which in turn improved soil nutrients. The enhancement of soil nutrients varied according to forms of manure returned to soil since nutrient components differed in various types of manure. The order of the concentration of total nitrogen and available phosphorus in waste droppings is sheep > pigs > cattle. Furthermore, the order of organic matter content of manure is sheep > pigs > cattle [48].

The environmental load ratio (ELR) is the ratio of possible non-renewable value to renewable energy value invested in the system. Compared to traditional agriculture systems, the ELR of crop–livestock systems decreased, mainly due to the reduction in the use of non-renewable resources by 6–38.5%. Organic fertilizers are the dominant renewable resources in the agro-ecosystem. In crop–livestock systems, sunlight could be fully utilized to promote the recycling of manure. The proportion of chemical fertilizers in non-renewable resources could drop by 4.09% as the application rate of organic materials (i.e., cow dung, bacterial residue) increases, which further optimizes the input structure of emergy. The ELR reduced, respectively, by 94.49–98.44% [19,43,44,45] and 22–84.66% [19,43,44,45] compared with single-cropping or breeding systems. The sustainable development index (SDI) is an indicator of the comprehensive performance capability of the evaluation system. The SDI of crop–livestock systems is generally 1.16–102.78 times that of traditional agricultural systems [43,44]. The degree of change in SDI depends on the ratio of net energy output rate to ELR. The transformation of agro-ecosystems to crop–livestock systems results in a more significant increase in SDI. Most crop–livestock systems can gain better ecological benefits. If the chain of agricultural and animal husbandry cycles is too long [45], production efficiency and benefits might drop due to accidents along certain links (Table 2). The mitigation level of greenhouse gas emission in different farming and animal husbandry cycle modes ranges from 2.72 × 103 kg CO_2_-eq/yr to 5.57 × 105 kg CO_2_-eq/yr [19,25].

#### 4.3.3. Social Benefits

The social benefits include improvements to the ecological environment, rural revitalization, and agricultural product quality and safety. Through effective treatment of fecal sewage generated by livestock farming, biogas biomass energy can be produced, and biogas slurry can be recycled as organic fertilizer resources, generating considerable ecological benefits [56]. By recycling rice, sorghum, and other crop straws, wastes can be turned into resources. Crop–livestock systems expand employment space for surplus rural labor forces [57]. The income source of the agricultural population can be guaranteed by the development of crop–livestock systems supported by the government since a stable working environment is provided, and rural revitalization is realized. The development of crop–livestock systems requires continuous innovation in science and technology, which promotes social development and improves social benefits.

### 4.4. Crop–Livestock System Stakeholders

The stakeholders of crop–livestock systems mainly include growers, farmers, and third-party institutions [58]. The individual characteristics of stakeholders, family conditions, and the surrounding environments are key drivers for crop–livestock systems.

Individual characteristics include education, age, and gender. The better education background they have and the younger they are, the more capable they will be of acquiring information and the more likely they are to accept the latest technologies concerning circular agriculture, such as biogas engineering and fermentation technology. In addition, men are more likely to use livestock and poultry manure resources than women [56].

The factors of family characteristics include annual family income, the concurrent employment state of family members, and the scale of cropping and breeding. Families with high annual income have more funds to invest in straw return or biogas projects and therefore are more likely to choose crop–livestock systems than those families with low annual income [36]. The opportunity cost of engaging in agricultural business activities improved with a higher degree of concurrent household employment, leading to less possibility for them to adopt circular farming techniques. Larger cropping or breeding farms tend to have higher straw and manure utilization rates because of higher scale benefits owners can obtain from crop–livestock systems [37].

External environmental factors include spatial proximity, neighbors’ opinion, credit support, and technical guidance. Spatial proximity affects the enthusiasm of farmers to return manure to the field and the coordination of crop–livestock systems by third-party agencies. When the spatial proximity is high, the cost of manure recycling is low, and farmers are more likely to return manure to the fields [38]. When the spatial proximity is low, the coordination of crop–livestock systems by the third-party institutions is high. The willingness of farmers to utilize straw is affected by factors such as neighbors’ opinions, credit support, and technical guidance [57].

## 5. Policy Recommendations

Farmers are the effective fulcrum for the development of crop–livestock systems. Taking farmers as the carrier, technologies such as straw silage, micro-storage, or producing pellet feed need to be promoted [59]. At the technical level, training farmers in nutrient management, including manure management, harmless treatment, storage, and use, and the appropriate selection according to local conditions in the integrated mode of planting and breeding, can help minimize the input of the combined planting and breeding system with increased output. Consequently, an efficient material circulation between crops and livestock can be achieved [60]. At the management level, guiding farmers to choose crop types rationally, planting areas, and number of livestock can help effectively reduce and control pests and prevent the occurrence of animal diseases [61].

As rational economic persons, the main consideration of farmers in their decision-making is whether they can gain better economic interests. On the one hand, preferential policies are given to farmers who are willing to participate in the combination of planting and breeding, and the loan conditions for farmers to transfer into land are relaxed [62]. On the other hand, corresponding subsidies will be given to farmers who have participated in the combination of planting and breeding, and farmers will be supported in the construction of equipment and facilities for the combination of planting and breeding [63]. In addition, the government should introduce other relevant policies and measures to promote the resource utilization of livestock and poultry manure on family farms, such as increasing the control of chemical fertilizers, promoting preferential subsidies for organic fertilizers to replace chemical fertilizers, and promoting the operation of market mechanisms [64].

In the process of integrating crop–livestock farming, the space for breeding land is still limited to some extent. Financial restraint is still a prominent problem facing farmers, and suitable mechanical breeding equipment is still insufficient [65]. In the future, we will improve land and financing policies to support the combination of planting and livestock farming and effectively improve the mechanization of the combination.

## 6. Conclusions

Based on the retrieved ISI Web of Science and CNKI data, research popularity, research methods, theme trends, and mainstream technology models of crop–livestock systems were sorted for China and elsewhere from 1981 to 2021. Based on a systematic review of ecological benefits, economic benefits, social benefits, and stakeholder perspectives of the crop–livestock systems, the following conclusions can be reached:(1)Regarding the process of research on crop–livestock systems, crop–livestock system research in other countries has gone through an initial development stage, stable growth stage, and rapid growth stage. Furthermore, the number of published papers has increased exponentially. Then, crop–livestock systems in China have gone through an initial development stage, rapid development stage, and fluctuation stage. Based on country-based publications, the top five countries are the United States, China, Germany, Australia, and Brazil.(2)Crop–livestock system models in other countries could be roughly divided into (i) low-input sustainable models and (ii) resource recycling models. The common technologies include the comprehensive utilization of straw and livestock manure. The mainstream models of crop–livestock systems in China are (i) “pig, biogas and fruit” eco-agriculture model in South China, (ii) “four-in-one” ecological model in North China, (iii) “five-support” ecological agriculture model in the northwest and iv) production and life cycle chain model in the plains.(3)The economic benefits of crop–livestock systems were reflected by the increase in output-input ratio and net income. The ecological benefits were reflected by the improvement in soil quality, reduction in environmental pollution, and mitigation of global warming. Then the social benefits were reflected by improvements in the ecological environment, employment opportunity, and agricultural product quality.(4)The stakeholders of crop–livestock systems are the growers, farmers, and third-party organizations. The individual characteristics of stakeholders, family conditions, and surrounding external environments are the key drivers of crop–livestock systems.(5)This study analyzed the impact of agro-pastoral and traditional agriculture on economic and ecological benefits; however, we did not combine the results of multiple studies in an effort to improve estimates of the size of the effect and resolve uncertainty due to disparities in reports, filling in the gaps in quantitative literature review methods.

## Figures and Tables

**Figure 2 ijerph-19-08563-f002:**
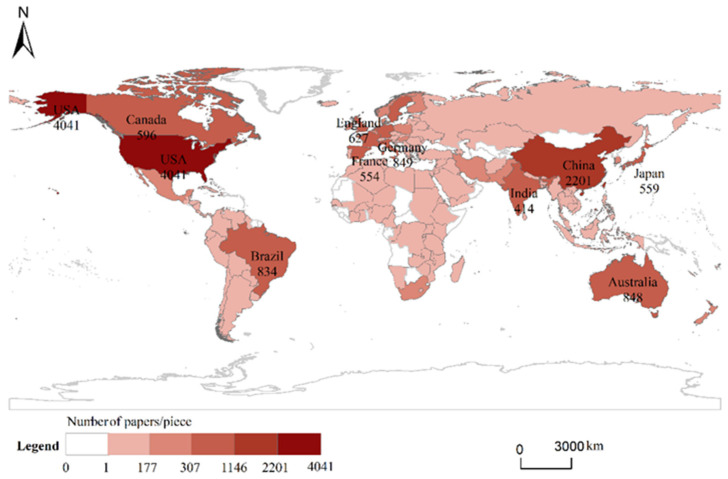
National distribution of crop–livestock system studies during the period 1981–2021.

**Figure 3 ijerph-19-08563-f003:**
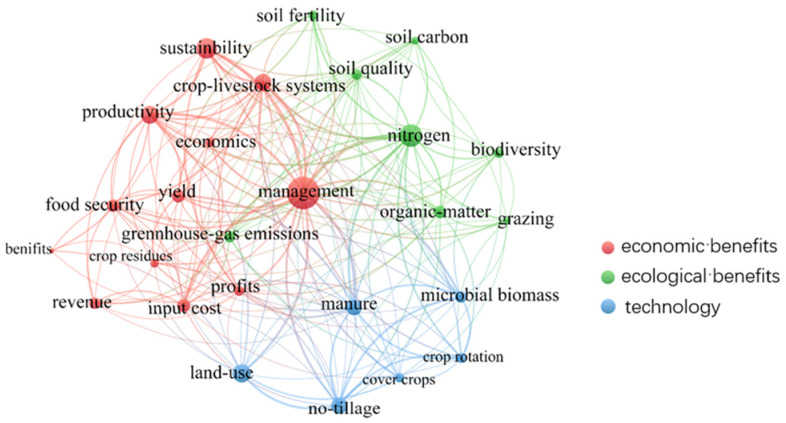
A map of the spread of keywords for the WOS database.

**Table 1 ijerph-19-08563-t001:** Methods to analyze the benefits of crop–livestock systems.

Category	Method Name	Function	Feature	Applicable Conditions	Related Software	Reference
Ecological benefit analysis	Emergy analysis	Emergy analysis via quantitative evaluation of indicators such as emergy self-sufficiency rate, environmental load rate and sustainable emergy development index of different farming and animal husbandry cycle modes.	Measure energy in social, ecological, and economic systems with unified emergy standard for quantitative comparative analysis in different systems.	Data involve four aspects: local renewable resources (wind, earth cycle, etc.), local non-renewable resources (soil loss, etc.), renewable resource purchase (labor, etc.) and non-renewable resource purchase (diesel, pesticide, etc.) data.	Excel, etc.	[30]
	Life cycle assessment	Quantitative analysis of ecological benefits of agro-pastoral cycles on global warming, environmental acidification, and soil toxicity	Evaluation based on the input-output inventory of materials, energy, and environmental emission during life cycle of product production or service.	An LCA inventory includes environmental emission data for life cycle of product or service.	eBlance, Gabi, SimaPro etc.	[31]
	Multi-objective optimization	Maximize problem solution for a set of objectives, consisting multiple linear functions under constraints	Solutions to multi-objective optimization problems, not a single isolated point but a collection of multiple optimal solutions.	Data includes three aspects: decision variables, objective functions, and constraints for constructing multi-objective optimization.	Matlab etc.	[32]
Economic benefit analysis	Data envelopment analysis	Relative efficiency of multiple decision-making units of output and input in crop–livestock system can be evaluated based on mathematical programming model.	No need to determine functional relationship in advance, non-subjective weighting, and can measure relative efficiency of multiple inputs and multiple outputs.	Data envelopment method does not need to determine the production function in advance, so it is more suitable for situations where uncontrollable factors have little effects on economic efficiency.	DEAP, Matlab, DEA, etc.	[33]
	Stochastic frontier analysis	After a certain amount of effective capital and labor are put into production, maximum productivity of farming and animal husbandry circular agriculture is calculated.	It is necessary to set production function form and solve efficiency through estimated parameters, to effectively distinguish between random errors and efficiency losses.	In establishing stochastic frontier analysis model to measure efficiency of agro-pastoral circular agriculture, it is necessary to determine appropriate production function.	Frontier, etc.	[34]
Social analysis	Linear regression model	Shows significant relationship between independent and dependent variables and strength of effects of multiple independent variables on a dependent variable, and quantitatively analyze driving factors of farmers’ participation in agriculture and animal husbandry cycle.	Model run is fast and has large amount of data, does not require cumbersome calculations, and can build understanding and explain each variable from coefficients.	There is a certain correlation between dependent and independent variables	SPSS, etc.	[35]

**Table 2 ijerph-19-08563-t002:** Economic and ecological benefits of crop–livestock systems.

Efficiency Type	Serial Number	Cycle Mode	Agricultural Output–Input Ratio	Net Income	Reference
				($/yr/ha)	
Economic benefits	1	Lychee–chicken	2.80	14,899.22	[38]
		Traditional independent lychees	5.41	7640.63	
	2	Cow–biogas–vegetables	1.25	40,969.16	[25]
		Traditional independent cattle breeding	1.11	16,960.35	
		Traditional independent vegetables	1.91	19,140.97	
		Fruit (grass)–pig–biogas–cellar five packages	4.02	23,367.36	
		Traditional independent pig farming	2.81	5670.36	
		Traditional independent planting of fruit trees	2.95	10,119.87	
		Mountain stereoscopic planting”	4.05	9215.99	
		Traditional independent chicken farming	1.21	792.5	
		Traditional independent walnuts	7.19	6598.40	
	3	Pig–biogas–forage, corn, kale	1.41	52,342.45	[45]
		Traditional independent pig farming	1.37	43,998.76	
	4	“Pig–biogas–pomegranate” pig raising subsystem	1.28	519,992	[44]
		Traditional independent pig	1.26	513,325	
		“Pig–biogas–pomegranate” pomegranate planting system	7.75	12,595.31	
		Traditional independent pomegranate cultivation	3.78	10,196.88	
	5	Pig–biogas–grain	2.55	255.8	[43]
		Traditional independent pig system	2.16	213.76	
	6	Rice–duck	1.98	650.6	[36]
		Traditional independent rice	1.73	310.61	
	7	Cow–corn/soybean	1.96	812.7	[40]
		Traditional independent cultivation of corn/soybean	1.75	1456.83	
		Traditional independent cattle farming	2.58	186.08	
	8	Beef–soybeans	1.78	674.17	[39]
		Traditional independent beef cattle	1.03	5.22	
		Traditional Independent soybeans	1.09	66.73	
		Increase soil organic matter (g/kg/yr)	Increase soil available phosphorus (mg/kg/yr)	Increase soil available potassium (mg/kg/yr)	
Ecological benefits	9	1.58	2.61		[46]
	10	2.28	7.55	87.85	[48]
	11	1.33	1.57	2.5	[42]
	12	0.73	1.3	23.79	[49]
	13	6.05	18.85	20.03	[50]
	14	4.75	17.92	20	[37]
	15	1.11	4.83	25.56	[51]
	16	1.15	1.64	1.6	[47]
			Environmental load rate (%)	Sustainability index	
	17	Crop–livestock system	3.23	0.37	[52]
		Traditional agriculture	5.14	0.2	
	18	Crop–livestock system	2.37	0.05	[19]
		Traditional agriculture	2.78	0.43	
	19	Crop–livestock system	3.88	3.76	[53]
		Traditional agriculture	5.9	2.9	
	20	Crop–livestock system	0.51	13.25	[45]
		Traditional agriculture	0.6	11.46	
	21	Crop–livestock system	1.43	1.22	[9]
		Traditional agriculture	1.94	0.92	
	22	Crop–livestock system	1.04	1.09	[30]
		Traditional agriculture	3.96	0.4	
	23	Crop–livestock system	0.07	42.55	[54]
		Traditional agriculture	2.22	0.41	
	24	Crop–livestock system	1.71	0.25	[55]
		Traditional agriculture	21.12	0.01	

## Data Availability

Available upon request.

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
