# Peer review of "Research Trends in Crop–Livestock Systems: A Bibliometric Review"

_ijerph, 2022, doi:10.3390/ijerph19148563_

Round 1

Reviewer 1 Report

The manuscript is well-written from the Introduction through the Methods. 

The manuscript needs expansion in the Conclusions respective to how Crop-Livestock Systems can be diffused  better to farmers/stakeholders. This inclusion would not only enhance the scholarship of the manuscript but potentially improve practices with Extension systems and/or enhance Crop-Livestock Systems adoption by targeted farmers/stakeholders. As an example, see the Conclusions identified in https://doi.org/10.3390/su131810295.

Author Response

Thank you for the comments,please see the attachment.

Reviewer 2 Report

I have serious concerns about the novelty of the paper as I am not convinced about the contribution of the paper in the literature. 

On the one side, the paper is about systematic review of the earlier literature but very short literature is quoted in the paper.

There is not a separate section of the literature review even.

Moreover, what can we conclude different from earlier literature?

Authors need to incorporate some novelty in the paper which should shed some light on a different aspect of the livestock sector. 

Author Response

Thank you for your comments, please see the attachment.

Reviewer 3 Report

To further improve the text, I suggest the following changes in the manuscript.
• Abstract: Abstract should be written in concise. I would suggest listing only some of the most important results to justify the implications and conclusions of the study.
• Keywords should not be included in the title. Please remove or substitute.
• The background of an introduction should be revised accordingly. This section must be upgraded with some latest references. I will suggest to read these articles and cite properly,

  • 10.3389/fenvs.2022.900193
  • 10.3389/fmats.2022.864254
  • 10.15244/pjoes/134292

https://link.springer.com/article/10.1007/s11356-021-16167-5

https://link.springer.com/article/10.1007/s11356-021-12867-0

• The introduction is very good. It doesn't reflect the goal; please rewrite it again, 
• Objectives of this study must be included at end of introduction part more clearly. 
• What is contribution of this work to existing literature? 
• It has been observed that the authors have used old references and ignored the latest studies. So it is suggested to add recent references. Please check reference section some references are missing.
• The policy implications also required elaboration. The implications should go along with the results and the course of action should be discussed in this part.
• Research Limitation and recommendations must be included in conclusion part.
• In some places, some grammatical errors are found that need to be fixed.

Author Response

(The authors gave the same response as above.)

Round 2

Reviewer 1 Report

The expanded literature review and expansion in the conclusions including the added Policy recommendations enhanced the manuscript. 

However, the inclusions in the Data source to identify the keyword search procedures were necessary for improvement. 

Author Response

Thank you for the comments. We improved the keyword search procedures.

Reviewer 2 Report

The authors improved much.

Author Response

Thank you for the comments.

Reviewer 3 Report

accept 

Author Response

Thank you.